# A Single Dose of AC102 Reverts Tinnitus by Restoring Ribbon Synapses in Noise-Exposed Mongolian Gerbils

**DOI:** 10.3390/ijms26115124

**Published:** 2025-05-27

**Authors:** Konstantin Tziridis, Jwan Rasheed, Monika Kwiatkowska, Matthew Wright, Reimar Schlingensiepen

**Affiliations:** 1Experimental Otolaryngology, ENT Clinic Head and Neck Surgery, University Hospital Erlangen, 91054 Erlangen, Germany; 2AudioCure Pharma GmbH, 10115 Berlin, Germany; mk@audiocure.com (M.K.);; 3Wright Pharma Consult, CH-4053 Basel, Switzerland

**Keywords:** hearing recovery, acoustic trauma, GPIAS, ABR, tinnitus, pharmacological intervention, synaptopathy

## Abstract

A single intratympanic application of the small-molecule drug AC102 was previously shown to promote significant recovery of hearing thresholds in a noise-induced hearing loss model in guinea pigs. Here, we report the effects of AC102 to revert synaptopathy of inner hair cells (IHCs) and behavioral signs of tinnitus in Mongolian gerbils following mild noise trauma. This experimental protocol led to minor hearing threshold shifts with no loss of auditory hair cells (HCs) but induced synaptopathy and a sustained and significant tinnitus percept. Treatment by intratympanic application of AC102 was evaluated in two protocols: 1. three weekly injections or 2. a single application. We evaluated hearing threshold changes using the auditory brainstem response (ABR) and the development of a tinnitus percept using the gap prepulse inhibition of acoustic startle (GPIAS) behavioral response. The number of IHC ribbon synapses along the cochlear frequency map were counted by immunostaining for the synaptic ribbon protein carboxy-terminal binding protein 2 (CTBP2). AC102 strongly and significantly reduced behavioral signs of tinnitus, as reflected by altered GPIAS. Noise-induced loss of IHC ribbon synapses was significantly reduced by AC102 compared to vehicle-treated ears. These results demonstrate that a single application of AC102 restores ribbon synapses following mild noise trauma thereby promoting recovery from tinnitus-related behavioral responses in vivo.

## 1. Introduction

Hearing deficits may not be detected by significant changes in pure tone audiograms but rather by reduced speech recognition. This type of hearing loss (HL) has been referred to as hidden hearing loss (HHL) and often remains undetected in patients [1,2]. Patients with HHL typically report difficulty hearing, especially in background noise. HHL occurs due to cochlear damage that affects neural communication through synaptic transmission between hair cells (HCs) and spiral ganglion neurons (SGNs), thus leading to reduced central auditory processing [3].

Both HL and HHL often lead to secondary impairments such as subjective tinnitus [4,5,6,7]. Beyond the negative impact on quality of life for the individual, tinnitus has a large socioeconomic impact related to a high incidence of psychological comorbidities that require significant mental health support [8].

In many cases, the tinnitus phantom percept is perceived as more debilitating than the HL itself [9]. Current approaches for the treatment of tinnitus provide minimal benefit. The most successful therapies rely on psychosomatic coping strategies [10] and cognitive behavioral or tinnitus retraining therapies [11,12]. Physiological approaches, which are rarely used, include deep brain or vagus nerve stimulation [13]. Non-invasive approaches include notched music [14], desynchronizing acoustic stimulation [15], or masking the percept with noise [16]. Hearing aids are provided to patients with significant HL that has been detected by pure tone audiometry. In contrast, no treatments are available to patients presenting only with tinnitus, leaving these individuals without treatment options [17].

An alternative to technical devices is to target underlying pathologies with therapeutic interventions. One potentially attractive approach is to reverse the synaptopathy of cochlear IHCs to prevent progression to auditory neuropathy [18]. In the Mongolian gerbil noise-induced tinnitus model, synaptopathy of IHCs has been shown to be causally linked to the development of a tinnitus percept [2]. Thus, reversing the synaptopathy of IHCs may be of benefit in the treatment of tinnitus.

AC102 is a newly developed small molecule pyridoindole drug under clinical development for HL. In a noise-induced HL model in guinea pigs, AC102 reversed IHC synaptopathy, suggesting benefit in reconnecting auditory neurons to sensory target cells [19]. Similar benefit was also observed in studies of cochlear implant-induced HL [20,21]. AC102 is currently under evaluation in a randomized phase 2 clinical trial in patients with sudden sensorineural HL (EudraCT No. 2024-513658-31-00).

The protocol employed in the present study employs mild noise exposure that induces transient elevation of hearing thresholds with IHC synaptopathy in the absence of HC loss [2]. The changes in the cochlea during the weeks following noise are often accompanied by behavioral changes that can be reflected by altered gap prepulse inhibition of the acoustic startle (GPIAS) response [22,23]. Altered GPIAS has been correlated to a tinnitus percept in the auditory cortex. In contrast to the rapid recovery of hearing thresholds after noise exposure, the tinnitus percept is durable and can be detected for up to 16 weeks. This suggests that chronic synaptopathy is established early following noise exposure.

In this study, we evaluated the hypothesis that AC102 delivered into the round window niche of Mongolian gerbils may revert the development of noise-induced synaptopathy, leading to recovery from tinnitus.

## 2. Results

### 2.1. Effects of AC102 on Hearing Thresholds

Prior to randomization, baseline hearing thresholds were determined by measuring auditory brainstem responses (ABRs) in all experimental groups. No significant differences in baseline hearing values at the measured frequencies between the intended treatment and control groups were detected (Appendix A).

In the triple application paradigm, hearing thresholds were significantly elevated directly after noise exposure (Figure 1). Threshold shifts across the frequencies ranged from 4 dB at 1 kHz to 20 dB at 4 kHz, with no differences between AC102- and vehicle-treated ears. By day 7, hearing thresholds had significantly recovered in both groups with no significant differences noted between treated and control animals. Two and five weeks after noise exposure, the improvement in hearing thresholds by AC102 was significantly greater than in the controls with the 4 kHz frequency being the most strongly affected. Notably, at the five-week timepoint, hearing thresholds had recovered to within 5 dB of baseline in the AC102 group with a significant shift still evident for vehicle-treated ears at 4 kHz (*p* < 0.001).

In the single application protocol, hearing thresholds were significantly elevated shortly after noise exposure (Figure 2). The magnitude and frequency-dependent threshold shifts mirrored the values found in the triple application protocol, ranging from 3 dB at 1 kHz to 23 dB at 4 kHz, with no differences between AC102- and vehicle-treated ears. By day 7, thresholds had significantly recovered in both groups. Two and five weeks after noise exposure, hearing thresholds continued to improve in the AC102 group to a significantly greater extent vs. the control group. By the last observation timepoint (day 35), hearing thresholds had completely returned to baseline across all frequencies in the AC102 group while the hearing thresholds in the control animals remained significantly elevated except for the 8 kHz frequency.

### 2.2. Effects of AC102 on Tinnitus-Related Behavior

As described in the Methods Section, effect sizes were calculated to determine if an animal showed a behavioral correlation of tinnitus at a specific frequency. The log-normalized prepulse inhibition (PPI) of post- and pre-measurements was assessed via Student’s *t*-test. A frequency was defined as tinnitus-positive (T+) if a significant decrease in PPI post- relative to pre-measurement was detected. Frequencies for which no change in PPI was detected were defined as tinnitus-negative (T−). The median effect sizes for both AC102- and vehicle-treated animals across all frequencies and all three timepoints of the triple protocol are given in Appendix A. For comparison of the AC102- vs. vehicle-treated groups, the T+ and T− frequencies were scored and compared using a chi^2^ test. We found significantly fewer frequencies with behavioral signs of tinnitus (*p* = 0.001) in the triple AC102-treated animals than in the vehicle group (AC102: T− N = 152, T+ N = 8; Vehicle: T− N = 160, T+ N = 30).

The number of animals with or without at least one tinnitus frequency at the timepoints following noise exposure is shown in Figure 3. The triple AC102-treated group showed significantly fewer animals with behavioral signs of tinnitus at each timepoint (*p* = 0.02 to *p* = 0.04). The percentage of animals in the two groups exhibiting tinnitus percepts were as follows: week 1: AC102 18% (2/11), vehicle 67% (8/12); week 2: AC102 18% (2/11), vehicle 58% (7/12); week 5: AC102 9% (1/11), vehicle 50% (6/12).

For the single application protocol, the resulting median effect sizes for both groups at the four timepoints are presented in Appendix A. The results were consistent with the triple application protocol. Significantly fewer frequencies scored positive for behavioral signs of tinnitus (*p* < 0.001) in AC102-treated animals vs. controls (AC102: T− N = 220, T+ N = 20; vehicle: T− N = 70, T+ N = 138).

The number of T+ and T− animals at different timepoints is presented in Figure 4. The single application AC102-treated group showed significantly fewer animals with behavioral signs of tinnitus at each timepoint (*p* < 0.001). It is important to note that differences between the protocols allowed for GPIAS to be conducted earlier in the single vs. triple application experiments. The addition of this earlier timepoint (day 3–4) revealed that an equally high percentage of animals in the AC102 and vehicle groups exhibited a tinnitus percept shortly following noise exposure (AC102 60% (9/15); vehicle 62% (8/13)). This earlier timepoint could not be assessed in the triple application protocol.

AC102 was able to promote reversion of tinnitus-related behavior in our animal model. At week 5, only one of fifteen animals treated by a single dose of AC102 scored “tinnitus positive”. In contrast, no animals recovered in the vehicle group, with eleven of thirteen scoring “tinnitus positive” at week 5. The percentage of animals in the AC102 and vehicle group exhibiting altered GPIAS were as follows: day 3–4: AC102 60% (9/15), vehicle 62% (8/13); week 1: AC102 20% (3/15), vehicle 85% (11/13); week 2: AC102 13% (2/15), vehicle 77% (10/13); week 5: AC102 7% (1/15), vehicle 85% (11/13).

### 2.3. Effects of AC102 on IHC Ribbon Synapses

Five weeks after noise exposure, animals were sacrificed, and immunohistochemistry was performed to quantify the number of ribbon synapses. Eleven sites along the tonotopic axis were selected and the synapses associated with 15 IHCs were counted (examples shown in Figure 5A). The average synapse number per cell was calculated for each cochlear location. Two-factorial ANOVAs of the synapse number differences with the factors *frequency* and *group* were performed independently for each dosing protocol.

In the triple application protocol (Figure 5B), IHCs had significantly greater numbers of ribbon synapses in AC102 ears in the *group* analysis (0.60 ± 0.23 syn/IHC; *t*-test vs. 0: *p* < 0.001). In contrast, a significantly lower number of ribbon synapses per IHC were present in the vehicle ears (−0.54 ± 0.25 syn/IHC; *t*-test vs. 0: *p* < 0.001). No *frequency* dependence of differences in ribbon synapse number, averaged over both groups, and interaction between the factors, was found. This indicates a frequency-independent preservation of ribbon synapses by AC102.

In the single application protocol (Figure 5C), there was a significantly greater ribbon synapse loss in the vehicle group (−0.91 ± 0.15 syn/IHC; *t*-test vs. 0: *p* < 0.001) compared to the AC102 group (−0.21 ± 0.14 syn/IHC; *t*-test vs. 0: *p* = 0.14). Synapse loss was frequency-dependent, with the greatest decrease in synapses in the frequency domain of 2 to 4 kHz (2 kHz: −1.66 ± 0.34 syn/IHC; 2.8 kHz: −1.78 ± 0.33 syn/IHC; 4 kHz: −1.56 ± 0.34 syn/IHC; Tukey post hoc tests: *p* < 0.05). We did not find any interaction between the factors (*p* = 0.13), indicating a parallel shift of the higher synapse number loss in the vehicle group compared to the AC102 group.

## 3. Discussion

Tinnitus can be a consequence of continuous or acute noise exposure that permanently disrupts synaptic connections between sensory IHCs and cochlear nerve fibers [1]. Animal studies have shown that noise exposure can result in synaptopathy, even if threshold elevation is transient or hidden [2,4]. The loss of peripheral synaptic terminals and reduction in neural activity due to subsequent neurodegeneration are, therefore, considered to be major contributors to the development of tinnitus and hyperacusis, the most common sensory comorbidities associated with sensorineural hearing loss (SNHL). Central auditory circuits may compensate by increasing synaptic gain to counter reduced neural signals from the cochlea [24]. However, reduced neural signaling to higher auditory centers due to IHC synaptopathy may predispose to tinnitus [25]. Future pharmacological therapies should be pursued focusing on treatment at the onset of cochlear deficits to counter HL and prevent tinnitus from becoming a chronic condition.

Previous studies have shown that the small pyridoindole molecule AC102 possesses both preventive and regenerative properties towards auditory neurons and synaptic connections in animal models of noise-induced HL and cochlear implantation [19,20,21]. Here, we have investigated AC102’s potential as a treatment for tinnitus. We employed in the well-characterized GPIAS model that assesses a noise-induced behavioral response, considered a surrogate for tinnitus.

We found that moderate noise exposure induced threshold shifts that recovered to almost baseline levels within five weeks in AC102-treated animals. In contrast, hearing thresholds remained significantly elevated in vehicle-treated animals, indicating mild chronic hearing impairment. Although this was observed for both experimental protocols, the effects of a single application of AC102 showed better hearing recovery compared to the triple application protocol. The differences may be due to several factors. It is possible that the technical procedures required for the three surgeries in the latter protocol may have interfered with recovery of the threshold shifts. There may also be an influence of animal variability. The two studies were performed on two separate cohorts. Of note, average hearing thresholds of the cohort employed in the triple application study were about 4 dB lower (*p* = 0.009) than for the cohort in the single application study. This factor may have altered the response to the effects of the acoustic trauma. Nevertheless, in both studies, AC102-treated animals showed a reduction in tinnitus-related behavior over time. This could be seen best in the single application protocol, where in both groups, about two-thirds of animals developed tinnitus-related behavioral alterations in the GPIAS response 3 to 4 days following noise exposure. While animals in the vehicle group did not recover during the five-week observation period (11/13 affected at 5 weeks), a single dose of AC102 reversed tinnitus-related GPIAS response in all but a single animal (1/15 affected at 5 weeks). Ribbon synapse counts further revealed a significant reduction in synaptopathy in the AC102 group compared to controls.

This study yielded proof of principle that a single application of AC102 into the middle ear is highly effective in reducing acute tinnitus-related behavioral responses in animals. The benefit of AC102 likely relates to its multimodal mechanism of action. AC102 was previously shown to promote recovery of hearing after noise or cochlear implant-induced HL through anti-inflammatory and anti-apoptotic mechanisms [19,20,21]. In those studies, a significant preservation of IHC ribbon synapses numbers along the entire cochlear tonotopic axis was found in AC102-treated ears. In addition, these prior studies reported significant neuroprotective effects of AC102 in the absence of IHC loss, as well as ability to promote neurite outgrowth. The reported observations may be explained by the effects of AC102 to increase cellular adenosine triphosphate (ATP), suggesting improved mitochondrial function, and to reduce reactive oxygen species (ROS) [19,21]. Both are important factors in the pathophysiology of SNHL and tinnitus.

Candidate otoprotective substances are often applied directly to the cochlear round window membrane to assure delivery of therapeutic drug concentrations and to reduce systemic side effects [26,27]. However, the optimization of drug formulations for effective absorption and distribution throughout the cochlea has been underappreciated. AC102 is formulated in a thermosensitive hydrogel that is liquid at room temperature and solidifies in the middle ear cavity at body temperature. The formulation has been extensively described in previous publications [28,29]. Notably, AC102 is effective over a wide frequency range, as shown in this and previous animal studies, indicating that the drug product distributes an effective concentration of AC102 up to the cochlear apex [19,21].

Despite the high medical need, attempts to develop new treatments for tinnitus have not progressed beyond mid-stage clinical trials. To date, no drug has been approved for the treatment of tinnitus by the FDA or EMA. Pharmacotherapy of acute tinnitus primarily aims to treat underlying HL. Steroids, delivered either systemically or locally to the middle ear, are the current standard of care for sudden SNHL [30,31,32]. However, recent results from the largest trial of steroids conducted to date (HODOKORT) did not provide evidence that steroids are beneficial in the treatment of sudden SNHL and, unexpectedly, demonstrated increased tinnitus percepts (frequency and loudness), poor recovery of hearing, and lower speech recognition in patients treated with high-dose steroids [33]. Thus, steroids appear to provide no benefit and may increase the incidence and severity of tinnitus. Prior to HODOKORT, only a handful of small studies had addressed the potential of steroids for tinnitus and had reported conflicting and inconclusive results [34].

In summary, we have shown that AC102 promotes recovery of tinnitus-related behavioral changes after mild noise exposure in the Mongolian gerbil. Recovery was coincident with the restoration of IHC ribbon synapses along the cochlear tonotopic axis. These data confirm AC102’s activity to reduce synaptopathy and improve hearing, consistent with previously reported data in models of sudden SNHL and cochlear implant trauma. Taken together, these data give a consistent picture of the therapeutic potential of AC102. In light of the controversial findings for corticosteroids, AC102 may address the unmet need for treatments for HL and its comorbidities, such as tinnitus. In a phase 1 study in healthy volunteers, intratympanic AC102 was found to be safe and well tolerated. AC102 is currently under evaluation in a European-wide phase 2 study in sudden SNHL patients (https://clinicaltrials.gov/study/NCT05776459, accessed on 22 May 2025). The trial is a double-blind, placebo-controlled study in approximately 200 patients with moderate, severe, or profound idiopathic sudden SNHL. AC102 is administered as a single intratympanic injection within five days of the onset of symptoms. The trial is being conducted at over 45 study sites across Europe, including Austria, Germany, the Czech Republic, the Netherlands, Poland, Serbia, and Ukraine.

## 4. Materials and Methods

### 4.1. Animals, Housing, and Ethics Statement

Fifty-one male Mongolian gerbils (Janvier, Le Genest-Saint-Isle, France) were used for these experiments. Animals were housed in standard cages (Bio A.S. Vent Light, Zoonlab, Emmendingen, Germany) in groups of 3 to 4, with free access to water and food at a room temperature of 20 to 25 °C under a 12 h/12 h dark/light cycle. At the beginning of the experiments, the animals were 12 weeks old and had been acclimated to their environment for at least 2 weeks. Animal care and use was approved by the state of Bavaria (Regierungspräsidium Unterfranken, Würzburg, Germany, No. 55.2.2-2532-2-914).

### 4.2. Experimental Protocol

The timelines for the experimental protocols are summarized in Figure 6. The GPIAS response behavioral paradigm was used to identify animals that developed tinnitus after noise exposure [23,35]. For determination of hearing thresholds, ABR measurements were conducted under anesthesia [36].

A monaural acoustic noise (115 dB SPL, 2 kHz, 75 min) was applied under anesthesia to induce a mild unilateral HL with the potential development of tinnitus (details below). ABR and GPIAS measurements were conducted before (baseline) and at three or four timepoints after noise exposure, depending on the application protocol (Figure 6). In the case of the triple application protocol (Figure 6A), surgery for drug application was performed on the noise-exposed ear directly after the ABR measurement, and again after the ABR measurements one and two weeks later. In the case of the single application protocol (Figure 6B), surgery and drug application were performed after the ABR measurement immediately following noise challenge. Animals were separated into two groups; one group received an AC102-containing gel formulation (AC102, triple application N = 11; single application N = 15) with a substance concentration of 12 mg/mL, and the other group received the vehicle gel formula without any drug (vehicle, triple application N = 13, single application N = 13). A volume of 10 µL of each formulation was applied into the round window niche by an UltraMicroPump 2 (World Precision Instruments, Friedberg, Germany) using a 50 µL syringe. Animals were sacrificed five weeks after noise exposure. Both cochleae were isolated, and immunochemistry was performed to determine the number of IHC ribbon synapses at 11 predefined locations.

#### 4.2.1. Monaural Acoustic Noise

Animals were placed under deep ketamine–xylocaine anesthesia (cf. above). One ear (pseudo-randomly selected) was plugged with foam (3M™ earplugs 1110, 3M, Neuss, Germany) to achieve at least 20 dB attenuation in the given frequency range. The animals were placed on a remote-controlled heating pad set to 37 °C with the non-plugged ear towards a speaker (CantonPlus XS 2, Canton Elektronik GmbH + Co. KG, Weilrod, Germany) at a distance of 10 cm. Acoustic noise (2 kHz, 115 dB SPL, 75 min) was only applied on that ear, usually the left. The contralateral, plugged ear served as control.

#### 4.2.2. Surgery

The animals were already under general anesthesia prior to surgery (cf. above and Figure 6). Under local anesthesia (lidocaine), the bulla of noise-exposed animals was surgically exposed sterilely on one side via a retroauricular incision and opened with a rose-head drill. Once the round window of the cochlea was clearly visible, a blunt cannula was used to apply the gel formulation with or without the drug. The contralateral ear received no treatment. The bulla was then closed with a medical-grade acrylic cement removable cover in the case of the triple application, or with a fixed seal in the case of the single application. The skin incision was then closed either “reopenable” or permanently, respectively. The entire surgical procedure required less than 30 min. An analgesic for pain (100 mg/kg metamizole) was administered postoperatively before animals regained consciousness, and for the first three days after the surgery.

#### 4.2.3. AC102 Gel Formulation

AC102 is a sterile suspension in a poloxamer-based thermosensitive hydrogel formulation. The formulation remains a liquid at ambient temperature and transitions to a gel at 37 °C, maintaining contact with the round window membrane to ensure cochlear distribution of the drug [19,21]. In the triple application protocol, animals received a total dose of 0.36 mg AC102, whereas animals of the single application protocol were dosed with 0.12 mg AC102.

#### 4.2.4. Behavioral Measurements

For the GPIAS measurements for tinnitus assessment, a custom-made open-source setup was used as described previously [37]. Animals were placed in an acrylic glass restrainer tube closed with a wire mesh at the front side and a cap at the back end and placed on a sensor platform fixed to a vibration-dampened table. Movements of the sensor platform were registered using a 3D accelerometer. Two loudspeakers were placed 10 cm in front of the animal. One loudspeaker presented a 115 dB SPL startle stimulus (Neo 25 S, SinusLive, Kaltenkirchen, Germany; noise burst 20 ms, flattened with 5 ms sin^2^ ramps) and the other the 60 dB SPL spectral noise background (CantonPlus XS 2). The spectral noise was centered at 1 to 16 kHz in octave steps with one octave bandwidth, with and without a gap of silence of 50 ms (flanked by 20 ms sin^2^ ramps, 10 ms complete silence) starting 100 ms before the startle stimulus.

Measurements were made at four or five timepoints following noise exposure (cf. Figure 6). Animals were given 15 min of adaptation in darkness in the tube. Prior to the measurement, five habituation stimuli were presented to “level” the startle response. After those stimuli, each experimental stimulus was repeated 30 times (15 with and 15 without gap), summing up to 120 stimuli (for details; cf. [37]). The entire procedure took approximately 30 min.

Evaluation of the GPIAS measurements was performed using custom-made Python 3.12 programs [37]. For the pre-noise measurements, the prepulse inhibition effect was controlled and found to reduce the startle amplitude by at least 20%. The GPIAS effect was quantified by calculating the mean from the full combinatorial log-normalized startle amplitudes as a response to gap and no gap pre-stimulus (for details, see [22]). Statistics on the mean GPIAS results were performed with Statistica 14 (TIBCO Software GmbH, Munich, Germany; cf. below).

#### 4.2.5. Brainstem Audiometry

For frequency-specific ABR measurements, animals were anesthetized with a ketamine–xylocaine solution (ketamine 500 mg/kg, xylazine 25 mg/kg). The animals were placed on a remote-controlled heating pad set to 37 °C. Three silver wires were used as electrodes and were placed subcutaneously, retroaural to the bulla of the tested ear (recording electrode), central between both ears (reference electrode), and at the base of the tail (ground electrode). Individual audiograms for both ears were obtained for stimulation frequencies between 1 and 16 kHz in octave steps for increasing stimulation intensities ranging from 0 to 90 dB SPL in 5 dB steps (6 ms duration with 2 ms sine square ramps) in a pseudo-random order. For each ear, stimulus, and intensity, 300 repetitions were presented, and the recorded responses were averaged. The complete measurement of one ear took around 30 min. The signal was recorded differentially between recording and reference electrode and filtered (bandpass filter 400 to 2000 Hz) via a Neuroamp 401 amplifier (JHM, Mainaschaff, Germany). ABR threshold analysis was performed automatically using previously described methods (cf. [36]).

#### 4.2.6. Immunohistochemistry

Immunohistochemistry was performed on cochlear whole-mounts as described previously [2]. Following completion of in-life behavioral and electrophysiological measurements, animals were sacrificed 5 weeks post-noise. The cochleae of both noise-exposed and contralateral control ears were extracted and fixed in 4% formaldehyde for 1 h. After decalcification in 0.1 M EDTA for 1–2 days, the cochlear turns were dissected into three segments. Cochlear segments were incubated overnight at 4 °C with a primary antibody against synaptic ribbon protein carboxy-terminal binding protein 2 (mouse anti-CTBP2 at 1:200; BD Transduction Labs, Heidelberg, Germany). A secondary antibody (donkey anti-mouse IgG conjugated with Cy3, 1:400, Dianova, Hamburg, Germany) was applied for 1 h at room temperature. Cochlear whole-mounts were then prepared for microscopy. The number of ribbon synapses associated with IHC were counted at 11 different sites along the tonotopic map, with 15 IHCs evaluated per site.

### 4.3. Data Evaluation and Statistical Analysis

Statistical analyses were performed with Statistica 14 after all data were obtained.

ABR threshold analyses were performed by an automated objective approach using the root mean square (RMS) values of the ABR amplitudes fitted with a hard sigmoid function using the background activity as the offset [36]. The mean threshold was set at the level of slope change of this hard sigmoid fit, independently for each frequency and for each timepoint. Data were controlled manually for non-valid fits and invalid analyses due to failure of the algorithm were discarded. This was the case for about 5% of all measurements, with the majority being threshold measurements immediately after the acoustic noise. This was expected, as the algorithm uses the neuronal background activity as reference for the determination of the evoked neuronal response threshold, which is often too high immediately after noise exposure. Threshold shifts were calculated by subtracting the post-noise from the pre-noise thresholds and evaluated by parametric tests. All statistical analyses were performed using unpaired Student’s *t*-test. Data are presented as mean ± standard error of the mean (SEM).

For the evaluation of GPIAS results, the means of the effect size of the individual log-normalized gap/no-gap amplitude responses of the different background stimuli were analyzed for each animal and each frequency [22]. Significant negative effect sizes (Student’s *t*-tests) at a given frequency indicate a tinnitus percept. Animals with a significant negative effect size for at least one frequency were scored as tinnitus positive (T+ group). The others were scored as tinnitus negative (T− group). Statistical comparisons between groups were performed using chi^2^ tests for the number of animals scoring T+ and T−, as well as for the number of frequencies affected in T+ animals.

The number of synapses per IHC was counted from the microscopic images by eye by one investigator blinded to the status of the cochlea (noise-exposed or contralateral control ear) or any other attribute of the animal. The frequency map of the cochlea from 500 Hz to 16 kHz in half octave steps was mapped using the Keyence BZ-II-Analyzer software Version 1.1 by measuring the relative distances beginning from the apex along the cochlea [2], according to the published cochlear frequency map for the Mongolian gerbil [38]. Full-focus microscopic images were captured for the IHC synapses of 11 distinct regions spanning inner spiral bundle to the nerve terminal in the supra-nuclear region, including all visible synaptic ribbons. Images were obtained with a fluorescent microscope BZ9000 (Keyence, Neu-Isenburg, Germany) with a 40× objective (0.6 numerical aperture) and visualized using an ImageJ Version 1.54 plugin. Labeled synapses were counted for groups of 15 IHCs in each of the 11 regions, according to the tonotopic map of the gerbil cochlea. For statistical analyses, factorial ANOVAs were performed. Data are presented as mean ± 95% confidence interval.

## Figures and Tables

**Figure 1 ijms-26-05124-f001:**
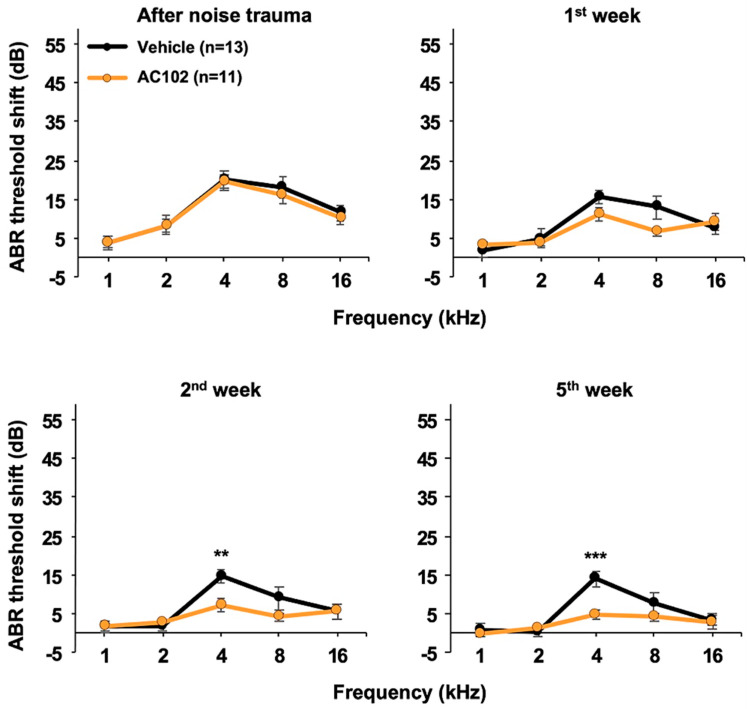
Development of hearing thresholds over time in the triple application protocol. Graphs depict the hearing threshold shifts over the frequency range vs. baseline values prior to noise challenge. Both AC102- and vehicle-treated ears exhibited modest threshold shifts ranging from 4 dB at 1 kHz to 20 dB at 4 kHz. The threshold shifts decreased over time reflecting recovery of hearing. By the end of the observation period the AC102 group had recovered to within 5 dB vs. baseline at all frequencies while the vehicle group exhibited a significant remaining threshold shift at 4 kHz (** *p* < 0.01; *** *p* < 0.001).

**Figure 2 ijms-26-05124-f002:**
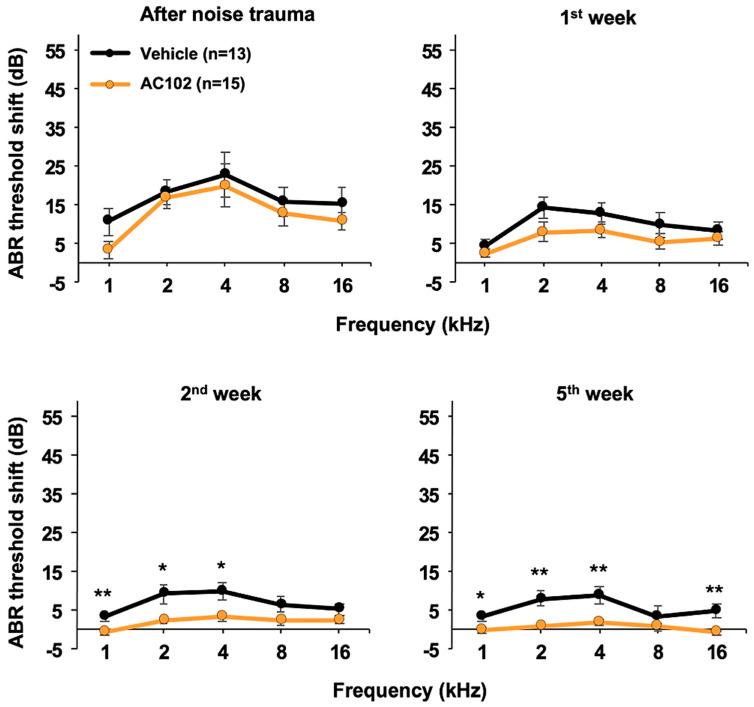
Development of hearing thresholds over time in the single application protocol. Graphs depict the hearing threshold shifts over the frequency range vs. baseline values prior to noise challenge. Both AC102- and vehicle-treated ears exhibited modest threshold shifts ranging from 3 dB at 1 kHz to 23 dB at 4 kHz. The threshold shifts decreased over time reflecting recovery of hearing. By day 35, hearing thresholds had completely returned to baseline across all frequencies in the AC102 group. In contrast, the vehicle animals had persistent threshold shifts ranging between about 5 and 10 dB (* *p* < 0.05; ** *p* < 0.01).

**Figure 3 ijms-26-05124-f003:**
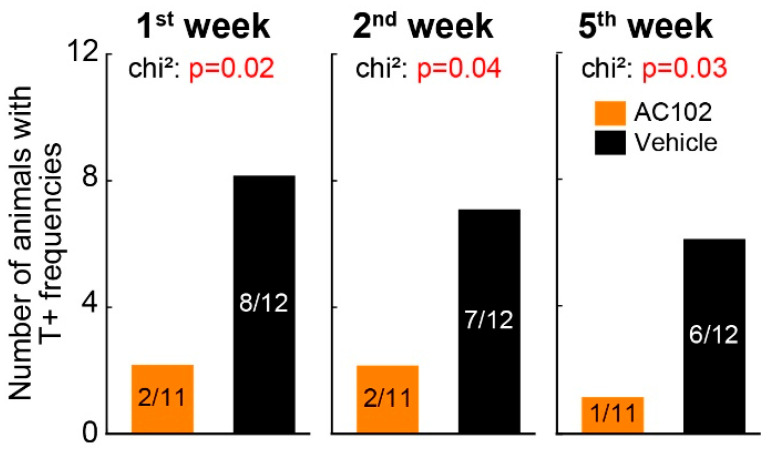
Tinnitus-related behavior reflected by altered GPIAS over time in the triple application protocol. Number of animals with at least one T+ frequency at different timepoints after noise exposure (panels). The AC102-treated animals showed significantly fewer behavioral signs of tinnitus than their vehicle-treated counterparts.

**Figure 4 ijms-26-05124-f004:**
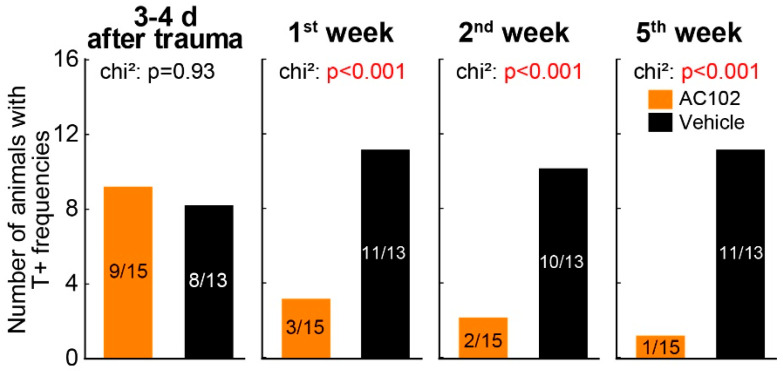
Tinnitus-related behavior reflected by altered GPIAS over time in the single application protocol. Number of animals with at least one T+ frequency at the different measurement times (panels). Except for the first post-noise measurement, AC102-treated animals showed significantly fewer behavioral signs of tinnitus vs. the control group.

**Figure 5 ijms-26-05124-f005:**
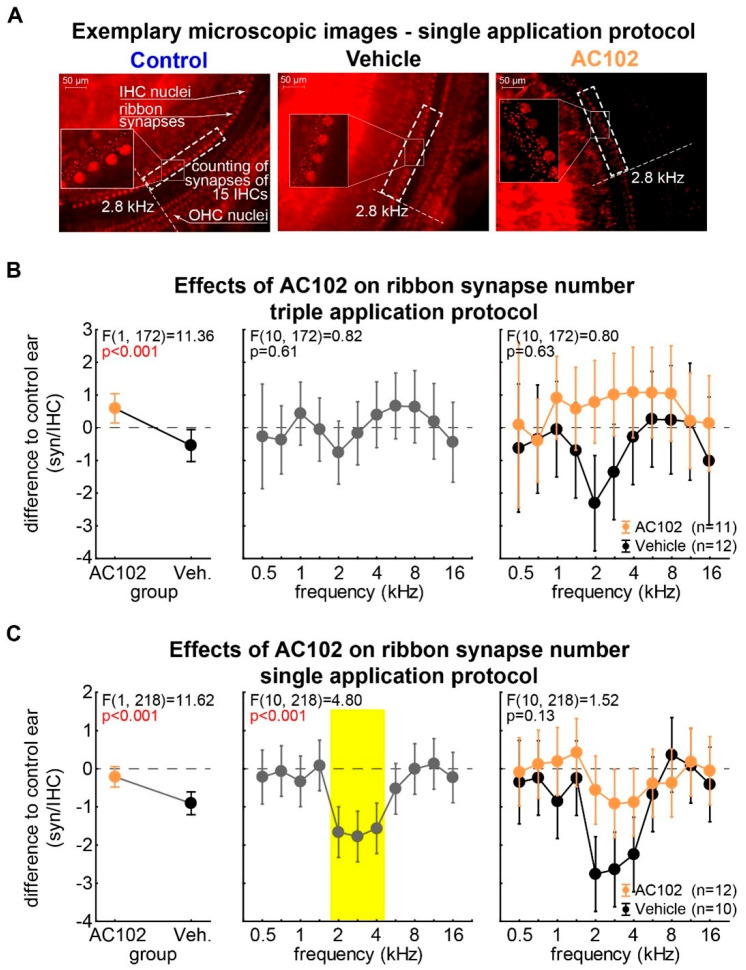
Effect of AC102 on IHC ribbon synapses. Change in the number of ribbon synapses per IHC in AC102- and vehicle-treated ears relative to normal hearing ears in both experimental protocols. (**A**) Representative micrographs of cochlear segments from control, vehicle, and AC102-treated ears stained for CTBP2 to detect synapses (see Section 4). The nuclei of inner (IHCs) and outer hair cells (OHCs) are visible (scale 50 µm). The number of ribbon synapses per IHC was counted for multiple segments along the tonotopic access, each segment consisting of 15 IHCs (white rectangles). The data shown here are for the segment corresponding to the 2.8 kHz frequency. Insets are higher magnification images of four IHCs and their ribbon synapses. (**B**,**C**) Results of 2-factorial ANOVA of the mean relative synapse differences between AC102- and vehicle-treated animals with the factors *group* and *frequency* (**B** triple and **C** single application protocol). **Left panels**: Results for the factor *group* averaged over all frequencies. AC102-treated ears show significantly more synapses/IHC than vehicle-treated ears in both experiments. **Center panels:** Results for the factor *frequency* for both groups combined. No differences were observed in the triple application protocol while significantly decreased synapse numbers were found in the single application protocol (yellow area; Tukey post hoc tests, *p* < 0.05). **Right panels**: Interaction plot of both factors. In both treatment protocols, the AC102-treated ears showed significantly more synapses per IHC compared to vehicle-treated controls.

**Figure 6 ijms-26-05124-f006:**
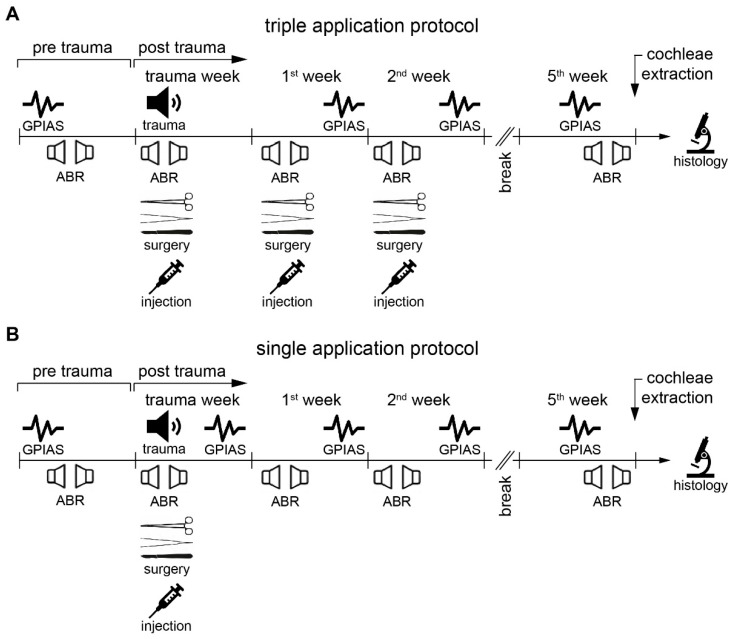
Schematic representation of experimental protocols. (**A**) Triple application protocol. GPIAS behavioral tests were performed on conscious animals. ABR, noise challenge, and surgery were performed under anesthesia. Retroauricular surgery and drug application into the round window niche was performed three times and one week apart. Immunofluorescence staining and microscopy for IHC synapse counting (histology) was performed after the sacrifice. (**B**) Single application protocol. Symbols and timing are as above, but only one drug application followed immediately after noise exposure. Note the additional GPIAS measurement during the noise trauma week.

## Data Availability

Data are available on request from the corresponding author with permission from the funders due to legal reasons.

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
