# Peer review of "A Single Dose of AC102 Reverts Tinnitus by Restoring Ribbon Synapses in Noise-Exposed Mongolian Gerbils"

_ijms, 2025, doi:10.3390/ijms26115124_

Round 1
Reviewer 1 Report
Comments and Suggestions for Authors
The authors investigated the application of AC102 in hidden hearing loss. Although AC102 did have the potential to prevent noise-induced damage, the overall results do not seem to support their claim for rescuing hidden hearing loss. The major concern is the model used for hidden hearing loss. Hidden hearing loss models should demonstrate synaptopathy without hearing threshold loss but Figure 1 shows permanent hearing threshold shifts after noise exposure. It would also be important to compare animals with and without noise exposure to evaluate the degree of synaptopathy in their model. In addition, it is interesting that triple treatment resulted in a slightly worse recovery, which is a concern that is worth discussing.
Another minor concern is about the effect of AC102 on tinnitus. The authors used GPIAS, which relies on sound input. As hearing thresholds shifted differently at different frequencies, it might be worthwhile to analyze responses at different frequencies.
Overall, the results show that AC102 rescued noise-induced hearing loss but will need to tune down their claims on hidden hearing loss or tinnitus. Some minor adjustments to the manuscript are listed below.
Ln 33-34, 34-37: citation needed.
Ln 48-52: citation needed to explain how hearing aid is relevant to tinnitus.
Ln 81: "data not shown" is less accepted now so would encourage them to present the data in the supplementary.
Ln 108-109: no data should be "negligible". I would suggest: minimal 5 dB threshold shift remained in the control group.
Ln 228: as explained earlier, this claim is not supported.
Ln 229-232: please specify the treatment - single or triple.
Ln 273-274: citation needed for corticosteroids.
Ln 413: "Student's" t test.
Author Response
please find our response attached in the file.

Reviewer 2 Report
Comments and Suggestions for Authors
The manuscript from Tziridis and co-authors entitled “A single dose of AC102 reverts tinnitus by restoring ribbon synapse in noise-exposed Mongolian gerbil” is generally nicely crafted but needs improvements as listed below. One major concern is stated last.
The abstract and body of the manuscript have statements that are not very scientific, sounding more like language intended for a lay audience rather than the scientific community. For example, line11 “almost full recovery”, line 13 “mild acoustic challenge.” line 24 “substantially reduced”, line 58, “novel small molecule”, lines 85 and 99, “moderately elevated”, line 103 “partially recovered”, line103 “a trend toward”, and line 162 “highly effective”. Provide statements that includes the data and eliminate the wishy-washy statements that lack a clear stance.
Figure 5, panel A, The three images are poor quality and need to be larger with better resolution.
The legend for figure 5 line 180, “exemplary” means “best of its kind”. I think you meant to say “An example of” or “For example”
Lines 220-221, “… aim on early…” is awkward phrasing. What does that mean?
Line 245 “… “… increased production of increased…” needs some help.
Line 275, provide more information about the clinical trial of AC102 and on line 340. Provide more details about AC102. This information will be of interest to readers.
What is exact composition of the poloxamer-based temperature sensitive hydrogel? Source/company/catalog number? How do you expect another laboratory to reproduce findings in your study if you conceal details?
Would you provide AC102 to other investigators if asked? If not, say so and see if the editor is agreeable? If so, say so. What do you intend to share with the scientific community if asked by investigators who want to reproduce your findings and continue on with their studies of AC102? There needs to be some honest statement about will and will not be shared with the scientific community in your manuscript. If reagents are proprietary and will not be provided for reproducibility studies, then be direct and say so and not have readers guessing.
Author Response

(The authors gave the same response as above.)

Round 2
Reviewer 1 Report
Comments and Suggestions for Authors
The authors have addressed all concerns. Please make sure there is no errors after accepting all changes for the final submission.
Author Response
The authors investigated the application of AC102 in hidden hearing loss. Although AC102 did have the potential to prevent noise-induced damage, the overall results do not seem to support their claim for rescuing hidden hearing loss. The major concern is the model used for hidden hearing loss. Hidden hearing loss models should demonstrate synaptopathy without hearing threshold loss, but Figure 1 shows permanent hearing threshold shifts after noise exposure.
Response: Reviewer 1 was satisfied with the revision.
Reviewer 2 Report
Comments and Suggestions for Authors
Authors have edited the manuscript appropriately. The revised version is much approved. In answer to a concern about not being able to reproduce the experiments because of proprietary issues, the authors stated in their response memo that "We acknowledge the reviewer’s concern regarding reproducibility. The material used in our study is based on a proprietary vehicle formulation developed by AudioCure Pharma. Due to need to protect intellectual property claims, the composition cannot be disclosed. However, the core vehicle formulation has been described in previous publications,including Harrop-Jones et al. (2016) or Salt et al. (2011) and results for the AC102 formulation have been published in other journals (Rommelspacher et al., 2024; Nieratschker et al., 2024). These references provide sufficient information to reproduce experiments".
I suggests adding a shorter version of this statement to the manuscript.
Author Response
Reviewer 2 requested addition of an abbreviated version of the statement:
"The material used in our study is based on a proprietary vehicle formulation developed by AudioCure Pharma. Due to need to protect intellectual property claims, the composition cannot be disclosed. However, the core vehicle formulation has been described in previous publications, including Harrop-Jones et al. (2016) or Salt et al. (2011) and results for the AC102 formulation have been published in other journals (Rommelspacher et al., 2024; Nieratschker et al., 2024). These references provide sufficient information to reproduce experiments".
In response we have added the following sentence to lines 271-272:
“The formulation has been extensively described in previous publications [ref,ref].”
We’ve added two references and updated the reference list:
Harrop-Jones A, Wang X, Fernandez R, et al. The Sustained-Exposure Dexamethasone Formulation OTO-104 Offers Effective Protection against Noise-Induced Hearing Loss. Audiol Neurootol. 2016;21(1):12-21. doi:10.1159/000441814
Salt AN, Hartsock J, Plontke S, LeBel C, Piu F. Distribution of dexamethasone and preservation of inner ear function following intratympanic delivery of a gel-based formulation. Audiol Neurootol. 2011;16(5):323-335. doi:10.1159/000322504